# Estimating Lab-Quake Source Parameters: Spectral Inversion from a Calibrated Acoustic System

**DOI:** 10.3390/s24175824

**Published:** 2024-09-07

**Authors:** Federico Pignalberi, Giacomo Mastella, Carolina Giorgetti, Marco Maria Scuderi

**Affiliations:** Dipartimento di Scienze della Terra, La Sapienza Università di Roma, 00185 Rome, Italy

**Keywords:** acoustic sensor calibration, acoustic emissions, source parameters, laboratory earthquakes

## Abstract

Laboratory acoustic emissions (AEs) serve as small-scale analogues to earthquakes, offering fundamental insights into seismic processes. To ensure accurate physical interpretations of AEs, rigorous calibration of the acoustic system is essential. In this paper, we present an empirical calibration technique that quantifies sensor response, instrumentation effects, and path characteristics into a single entity termed instrument apparatus response. Using a controlled seismic source with different steel balls, we retrieve the instrument apparatus response in the frequency domain under typical experimental conditions for various piezoelectric sensors (PZTs) arranged to simulate a three-component seismic station. Removing these responses from the raw AE spectra allows us to obtain calibrated AE source spectra, which are then effectively used to constrain the seismic AE source parameters. We apply this calibration method to acoustic emissions (AEs) generated during unstable stick-slip behavior of a quartz gouge in double direct shear experiments. The calibrated AEs range in magnitude from −7.1 to −6.4 and exhibit stress drops between 0.075 MPa and 4.29 MPa, consistent with earthquake scaling relation. This result highlights the strong similarities between AEs generated from frictional gouge experiments and natural earthquakes. Through this acoustic emission calibration, we gain physical insights into the seismic sources of laboratory AEs, enhancing our understanding of seismic rupture processes in fault gouge experiments.

## 1. Introduction

Seismic waves generated by rupture processes are a phenomenon observable across various scales, from microscopic grain breaking to macroscopic fault slip. When a rupture occurs, it instantaneously releases accumulated elastic energy. A portion of this energy is released as waves, which propagate through the surrounding medium. Rich in information, these seismic waves offer insights into both the rupture source and the properties of the medium through which they propagate. To date, the analysis of acoustic waves associated with earthquake rupture is the most important information at our disposal to understand the physics of earthquakes.

In laboratory settings, acoustic emissions (AEs) are detected as elastic waves generated by the rapid release of energy from breaking processes, resembling small-scale earthquakes. These AEs provide valuable insights into the physics of seismic events. Pioneering studies have used continuous AE monitoring to investigate rock deformation processes in various laboratory experiments [1,2,3,4,5,6], establishing AE activity as a reliable proxy of the stress state of laboratory faults. In experiments with bare rock surfaces or glass beads, AE activity follows the Gutenberg–Richter relation, where the b value varies with the stress state [7,8,9,10,11] and fault roughness [12]. Analyzing the spatial and temporal distributions of AE hypocenters preceding main stick–slip instabilities can provide fundamental insights into earthquake nucleation mechanisms [11,13,14,15,16,17]. By adapting seismological techniques to laboratory settings, comprehensive characterization of focal AE mechanisms [18,19] and source parameters [16,20,21] has become feasible. Comparing laboratory estimates of AE source parameters to seismological estimates of earthquake sources reveals a continuum in the scale relationship between seismic moment ($M_{0}$) and corner frequency ($f_{c}$), expressed as $M_{0}\propto f_{c}^{3}$ [22]. This relationship implies a consistent stress drop across a wide range of scales, encompassing laboratory experiments, underground in situ experiments, induced seismicity, and natural earthquakes [23,24,25,26]. The apparent scale-invariant nature of stress drop suggests that the underlying mechanisms of laboratory AEs and earthquakes are fundamentally similar. However, seismologically derived static stress-drop values are subject to considerable uncertainty due to assumptions about the employed source model and challenges in accurately determining corner frequency ($f_{c}$) [23,27,28,29]. This uncertainty results in a wide range of estimated stress drops spanning four orders of magnitude, from 0.01 MPa to 100 MPa, observed across the whole spectrum from small-magnitude ($M_{w}$~−8) laboratory AEs to large earthquakes exceeding magnitude 6 [20,21,23,30,31].

One notable and obvious difference between earthquakes and laboratory AEs lies in the source dimension, which influences the frequency content of the acoustic signals. Larger source ruptures typical of natural earthquakes generate seismic waves in the order of Hz to tens of Hz, whereas the smaller source ruptures of AEs, such as those observed in lab earthquakes, emit high-frequency seismic signals ranging from tens of kHz to MHz [32]. To circumvent this distinction, it is necessary to use piezoelectric transducers, commonly known as PZTs (Piezoceramic lead-Zirconate lead-Titanate) to detect these high-frequency signals in laboratory settings. Operating on principles similar to seismometers, PZTs are capable of effectively capturing seismic waves across a wide spectrum of frequencies, ranging from hundreds of Hertz to MHz. Despite their broad operational range, most PZTs operate in resonance, lacking a flat response throughout their operability frequency range. This characteristic implies that the recorded waveform is significantly influenced by the sensor type and the employed data acquisition system. Moreover, high-frequency waves experience significant geometrical attenuation and are sensitive to minor heterogeneities of the propagation medium. Consequently, an accurate physical interpretation of AEs requires a careful quantification of both instrumentation and path effects to effectively isolate the source components of the AEs.

Various experimental procedures have been employed to calibrate acoustic systems [33,34]. Among the most prevalent methods is the face-to-face technique, where two PZTs of the same type are used; one acts as a transmitter and the other as a receiver. Among others, the laser vibrometer method involves measuring the displacement of a sensor’s free surface in response to a perturbation. Another approach utilizes the generation of a known sharp pulse, such as the fracture of capillary glass or the dropping of a steel ball, which is then recorded by a receiver PZT.

In this work, inspired by McLaskey et al. (2015) [35] and McLaskey and Lockner (2016) [21], we describe a method of empirical absolute AE system calibration designed to quantify three elements, namely sensor response, instrumentation, and path effects, in a single measure defined as instrument apparatus response. To extract the instrument apparatus response up to 500 kHz, we use steel ball drops of varying sizes to generate a controlled source with varying magnitude. AE calibration techniques of such type have been previously applied in both direct shear and triaxial configurations on bare rock [21,35,36] and PMMA samples [16]. However, to the best of our knowledge, the application of such calibrations in fault gouge experiments has not been explored, despite the substantial body of experimental studies on earthquake mechanics using fault gouge as sample material. Extending the AE calibration method to fault gouge experiments is essential to derive valuable physical insights into the seismic nucleation and rupture processes generated in fault gouge.

In this article, we first provide a detailed description of the calibration procedure. Subsequently, we validate this procedure through direct application to AEs generated during stick–slip events in a double direct shear experiment, using quartz gouge as simulated fault gouge.

## 2. Calibration Test Setup

### 2.1. Deformation Apparatus

The calibration and test experiment are performed using a BRAVA2 biaxial apparatus at the laboratory of Rock Mechanics and Earthquake Physics at Sapienza University of Rome. The BRAVA2 biaxial apparatus consists of two orthogonal servo-controlled hydraulic pistons that apply loads to the sample. The horizontal piston is designed to provide fault normal stress, while the vertical piston provides the fault shear stress in the double direct shear configuration (Figure 1a). Each piston is equipped with a custom-built strain gauge load cell that is regularly calibrated and has an accuracy of 0.03 MPa. Displacement transducers, specifically Linear Variable Differential Transformers (LVDTs, Transtek Series 240 for the horizontal piston and Ametek Solatron S series for the vertical piston) are mounted between the load frame and the moving pistons to measure piston displacement with a resolution < 0.1 μm. Measurements of load point displacement are corrected for the elastic deformation of the experimental apparatus. We implement a non-linear elastic correction that accurately adjusts the load point displacement for the apparatus stretch at each specific applied force.

The Double Direct Shear (DDS) configuration consists of two layers of fault gouges sandwiched between the central block and two side forcing blocks (left inset Figure 10) made of steel. All the blocks are equipped with teeth to ensure mechanical coupling between the gouge and the blocks.

The hydraulic piston’s double-chamber movement is controlled through a MOOG D765 servo-valve, which uses electrical signals to move the spool and modulate the oil flow rate into the cylinder chamber. In this system, transducers such as load cell or LVDTs communicate their position or load to the control electronics in real time, which compare the feedback signal voltage with the desired voltage target and adjust the spool position through electrical feedback to eliminate any error and achieve the desired position or load.

Each subsystem’s control loop, whether for load or displacement, uses a PID (Proportional-Integral-Derivative) controller algorithm that is tuned to manage the system’s complexity. All output signals from load and displacement sensors are digitized using a multi-channel analog-to-digital converter (NI-9403) with a 24-bit/channel resolution and a maximum recording rate of 10 kHz.

The BRAVA2 control loops are implemented within an FPGA cRIO-9049 (Field Programmable Gate Array) integrated with a LabVIEW graphical user interface (GUI). This setup acquires data, controls the PID values, and regulates the input/output signals to the servo-valves in real time.

### 2.2. Acoustic System

The acoustic system consists of the following two main components: sensors (piezoelectric transducers, PZTs) and recording instrumentation. We use different types of PZTs from Physik Instrumente (PI) Ceramic—specifically, modified lead zirconate–lead titanate (PIC 255), the properties of which detailed in Table 1. Where $C_{55}^{E}$ and $C_{33}^{E}$ represents the elastic stiffness coefficients, indicating the material’s stiffness under constant electric field conditions; $S_{11E}$ is the elastic compliance coefficient, reflecting the ratio of relative deformation to mechanical stress; and $\sigma_{E}$ is Poisson’s ratio, which measures the ratio of transverse strain to axial strain when the material is subjected to stretching or compression.

Our sensors include both shear plates polarized perpendicular to the sensor surface and axial disks polarized parallel to the sensor surface. Due to PZT polarization, shear plates are more sensitive to the shear component (S), and axial plates are more sensitive to the compressional component (P). For this reason, we refer to these as S and P sensors, respectively.

Each PZT has a diameter (D) of 5 mm and a thickness (t) of 1 mm, resulting in the following resonant frequencies:

Thickness–shear resonance of S sensors:$$f_{r}=\frac{1}{2t}\sqrt{\frac{C_{55}^{E}}{\rho }}\approx 826\;\mathrm{kHz}$$

Axial resonance of P sensors:$$f_{r}=\frac{1}{2t}\sqrt{\frac{C_{33}^{E}}{\rho }\cdot \frac{\left(1-\sigma^{E}\right)}{\left(1+\sigma^{E}\right)\left(1-2\sigma^{E}\right)}}\approx 2.47\;\mathrm{MHz}$$

Radial resonance of P sensors:$$fr=\frac{2.08}{\pi D}\sqrt{\frac{1}{\rho S_{11}^{E}\left(1+\sigma^{E}\right)\left(1-\sigma^{E}\right)}}\approx 592\;\mathrm{kHz}$$

The PZTs are glued to the side blocks using 8331D Silver Conductive Epoxy Adhesive from MG Chemicals. A thin epoxy layer is applied to one surface of the PZT, which is subsequently compressed against the lateral block by a 500 g mass for a duration of 24 h. This procedure ensures robust mechanical coupling between the sensor and the forcing block, as well as electrical conductivity to ensure a good ground and limitation of the signal noise (Figure 1b).

Each side block hosts four PZTs arranged in a cross configuration (Figure 1b); axial sensors (P sensors) are positioned at the top and bottom, while transversal sensors (S_v_ and S_h_) are centered and oriented perpendicularly to capture the respective wave components. This configuration has the advantage of simulating a three-component seismic station on each side block.

The sensors within the side blocks are placed at a distance of 1.5 cm from the fault surface. The distance between the axial sensors and the central transversal sensors is 1.5 cm. The two central transversal sensors are located adjacent to each other, with a distance of 0.5 cm (Figure 1b).

Seismic waves are continuously recorded using three two-channel, fast-recording TiePie Handyscope HS5 digital oscilloscopes. The oscilloscopes are connected in series and guarantee real-time streaming of data. We use a recording rate of 6.25 MHz and a resolution of 16 bits, enabling high-precision analysis of the generated waveforms. The Handyscope recording parameters are controlled via a custom-made Python program that allows for choice of the recording frequency, resolution, and voltage range. All data are recorded in 1-s chunks to avoid buffering and data loss.

For all recordings during the calibration, a constant voltage range of 0.8 V (−0.8 to +0.8) is selected to avoid signal saturation, maximize resolution, and maintain uniformity across all recorded acoustic signals. No filters or amplifiers are applied during the recording process to preserve the integrity of the waveforms. To synchronize the acoustic and mechanical data, which are sampled at different frequencies and recorded on two distinct systems, we simultaneously record the shear force applied to the sample using oscilloscopes and the BRAVA2 system. We then cross-correlate the signals to ensure alignment.

### 2.3. Ball-Drop Setup and Procedure

For acoustic sensor calibration, we use 440 stainless-steel balls with diameters ranging from 0.5 mm to 15 mm (0.5, 1, 2, 3, 4, 5, 6, 7, 8, 9, 10, and 15 mm). The steel balls are dropped onto steel side blocks, each with a surface area of 5 × 5 cm and a thickness of 2.95 cm. An electromagnet fixed in a vertical position allows for the controlled release of the steel balls, ensuring zero kinetic energy upon ball release and only the influence of gravity on the ball impact (Figure 1a). However, smaller balls (1 mm and 0.5 mm in diameter) require manual dropping due to residual magnetization hindering their electromagnet release. Similarly, the excessive mass of the 15 mm ball precludes magnetization, necessitating a manual approach as well.

Our calibration setup differs slightly from the standard double direct shear (DDS) configuration used in standard shear test experiments [37,38,39]. Indeed, in our calibration setup, the simulated fault gouge is placed directly between the two side blocks without using the central block. We use a 3 mm thick gouge layer of MinUSil 10 (U.S. Silica Co.) composed of 99.5% SiO_2_ and trace amounts of metal oxides, with a medium grain size of 3.4 µm. The assembly is then placed in the BRAVA2 apparatus under normal load. The application of the normal load is fundamental to take into account the noise generated by the vibration transmitted from the hydraulic power supply that is present during the experiment under any boundary stress condition.

During the calibration test, the procedure begins by applying a starting normal load of 12 MPa, followed by a compaction phase of the gouge material for 30 min. Subsequently, from a fixed drop height of 11.5 cm, five ball drops are performed for each ball diameter on each side block. Afterwards, we increase the normal stress to 50 MPa and repeat the ball-drop procedure. For each steel ball drop, rebounds are carefully monitored to ensure they occurred on the same block as the first impact in order to calculate the rebound height necessary to correct the force imposed by the steel ball (see theoretical formulation section).

## 3. Theoretical Framework

AE signals capture the ground motion on the material surface resulting from mechanical disturbances or localized rupture processes that occur within the material. These ground motion signals are modified by the recording system and converted into electrical signals by the PZT. Mathematically, the system’s response can be represented as a linear system as follows:(1)s(x,t)=L[u(x,t)]
wherein the input function (u(x,t)) is transformed into the output function (s(x,t)), representing the electrical signal output of the ground motion, via a transfer function (*L*) associated with the AE recording system. Within this linear framework, the transfer function (*L*) represents how the signal is transferred throughout the system and includes all the system components that may alter the signal. These components include sensor type, propagation material, coupling sensor material, cables, amplifiers (if present), and filters (if present). Each component influences the system’s overall response; thus, ideally, the effect of each component on the transfer function (*L*) can be individually discerned. However, given the complexity and practical difficulties associated with this approach, it is conventional practice to combine all these elements into a singular comprehensive entity defined as the instrument apparatus response (i(t)).

Considering the instrument apparatus response as the transfer function, the AE recording system can be represented by the following equation:(2)s(x,t)=f(x,t)∗i(t)
where the sensor output (s(x,t)) is the convolution (∗) of the input signal (f(x,t)) with the instrument apparatus response (i(t)) (Figure 2).

Consequently, the calibration of the acoustic emission system involves the determination of the i(t) function. For the sake of simplicity, we operate in the frequency domain, where convolution simplifies to multiplication. We compute the instrument apparatus response in the frequency domain as follows (Figure 2):(3)$$I\left(\omega \right)=\frac{S(\omega )}{F(\omega )}$$
where capital letters represent the Fourier transforms of the respective components. The instrument apparatus response (I(ω)) is retrieved by employing the empirical Green function (eGf) method [40]. This method assumes that the recorded waveform includes information about the source, wave propagation through the material, and the characteristics of the recording system. Such a technique is commonly used in seismology to study seismic sources by using smaller, co-located earthquakes to eliminate path and site effects from larger target earthquakes [41,42,43,44].

For earthquakes occurring at the same location, we can assume that the wave travel path and recording conditions are the same, allowing for the removal of these common effects and isolation of the source. In our controlled laboratory settings, we systematically generate co-located seismic events of varying magnitudes by dropping steel balls of different diameters from a fixed height onto the same point of impact.

To retrieve the instrument apparatus response, we require the signal output (S(x,ω)), which is recorded by the oscilloscope, and the source function (F(x,ω)), which is responsible for producing the mechanical disturbance (Equation (Equation 3)). Following previous studies [35,45], we use Hertzian theory [34,46] to model the impulsive function of the steel ball drop. Under this theoretical framework, we can compute the maximum force impressed by the ball impact ($f_{\max}$) and the contact time ($t_{c}$) using the following equations:(4)$$\delta_{i}=\frac{1-\mu_{i}^{2}}{\pi E_{i}}$$
(5)$$t_{c}=4.53{\left(4\rho_{1}\pi \frac{\left(\delta_{1}+\delta_{2}\right)}{3}\right)}^{\frac{2}{5}}R_{1}v_{0}^{-\frac{1}{5}}$$
(6)$$f_{\max}=1.917\rho_{1}^{\frac{3}{5}}{\left(\delta_{1}+\delta_{2}\right)}^{-\frac{2}{5}}R_{1}^{2}v_{0}^{\frac{6}{5}}$$
where $\delta_{i}$ represents a factor influencing the elastic deformation between two contacting bodies, Poisson’s ratio (ν) describes the relationship between transverse contraction and longitudinal extension, and Young’s modulus (*E*) measures the material stiffness. Subscripts 1 and 2 denote the ball and the test specimen, respectively. Therefore, $t_{c}$ and $f_{\max}$ can be calculated by determining the density ($\rho_{1}$), the radius ($R_{1}$), and the impact velocity ($v_{0}$) of the steel ball.

Since the ball impact is not completely elastic, we must correct $f_{\max}$ for ball rebound ($f_{\mathrm{reb}}$) as follows:(7)$$f_{\mathrm{reb}}=\frac{1+\sqrt{\frac{h_{\mathrm{reb}}}{h_{0}}}}{2}f_{\max}$$
where $h_{0}$ is the initial drop height and $h_{\mathrm{reb}}$ is the first rebound height, which is calculated as follows:(8)$$h_{\mathrm{reb}}=0.5g{\left(\frac{t_{\mathrm{reb}}}{2}\right)}^{2}$$
where *g* is the gravitational acceleration and $t_{\mathrm{reb}}$ is the time between the first touch and the first rebound, which is calculated by manually determining the arrival times of the first touch and the first rebound (Figure 1c). Therefore, the corrected force results are expressed as follows:(9)$$f_{\max\;\mathrm{corr}}=f_{\max}-f_{\mathrm{reb}}$$

After determining $t_{c}$ and $f_{\max\;\mathrm{corr}}$, we can compute the force–time function (f(t)) using the following relation:(10)$$f\left(t\right)=\left\{\begin{array}{cc} f_{\max}\sin{\left(\frac{\pi t}{t_{c}}\right)}^{3/2} & \mathrm{for}\;0\leq t\leq t_{c}\\ 0 & \mathrm{otherwise} \end{array}\right.$$

By integrating the force–time function (f(t)) from 0 to $t_{c}$, we derive the change in momentum (Δp). This value is essential in correcting the spectrum of ball drops and computing I(ω) using Equation (Equation 3). Specifically, for frequencies of $f<f_{c}$ (where the spectrum is flat), F(ω)=Δp. In this way, the raw spectrum is corrected for the change in momentum by dividing S(ω) by Δp. Consequently, for $f<f_{c}$, the corrected spectra of all ball drops collapse into a single curve that represents the instrument apparatus response I(ω) (see Section 4). For a more detailed description of this theoretical framework, we refer to McLaskey and Glaser [45], McLaskey et al. [35], and McLaskey and Lockner [21].

## 4. Absolute AE System Calibration

### 4.1. Theoretical Force–Time Function

Using Hertzian theory, we derive the theoretical force–time function for all the performed ball drops. The derived values are listed in Table 2, where $\Omega_{0}$ represents the spectral amplitude of the flat portion in the Fourier spectrum and $f_{c}$ corresponds to $1/T_{c}$ (Figure 3a). Fourier transform (FFT) spectra of the theoretical force–time functions clearly show that larger balls (with higher mass) are characterized by larger $\Omega_{0}$ values, corresponding to a higher moment and a smaller corner frequency ($f_{c}$) (Figure 3b).

### 4.2. Waveform Analysis

For each ball drop, we focus only on AEs generated by the first pulse. We manually pick the wave arrival time and select a portion of the waveform corresponding to a window of 24,000 points (equivalent to 4 ms), starting 500 points before the arrival time of the first pulse (Figure 4a). Following McLaskey and Lockner (2016) [21], this window size is sufficient to obtain good spectral estimates with 5 kHz as a lower limit ($f_{\min}=20/T_{\mathrm{wind}}$). To reduce spectral leakage and minimize aliasing effects, we taper the waveform using a Tuckey window (Figure 4a). Such a window provides a smooth transition to zero within the 500 first and last points, preserving the integrity of the waveform. To enhance spectral resolution at lower frequencies, we apply zero padding, extending the waveform window to 120,000 points (Figure 4b). We carefully apply the same windowing, tapering, and zero padding to the first noise window recorded before the arrival of the AEs (Figure 4c). The determination of the noise for each AE is essential to determine waveform spectra with a sufficient signal-to-noise ratio (SNR).

### 4.3. Spectral Analysis

We compute the fast Fourier transform (FFT) for both the AE and associated noise. To reduce spectral noise, we bin the spectra and calculate the median value within each bin (Figure 3a and Figure 4d,e). We use 180 logarithmically spaced bins across a frequency range of 100 Hz to 3 MHz. To obtain a more stable spectral estimate for each ball size, we use the median of the binned spectra of the five drop tests performed for each ball size (Figure 5). Spectral analysis of ball drops of the same diameter shows high consistency at frequencies below $f_{c}$. However, at higher frequencies, noise contributes to small discrepancies between the spectra. To obtain robust spectral estimates, we compare the waveform and noise spectrum and use only the portion of the spectrum with a signal-to-noise ratio (SNR) > 10 dB (Figure 6a).

The raw spectra for all ball drops are shown in Figure 6. From the raw spectra, it is challenging to discern the corner frequencies, since the sensor response may show an apparent corner frequency—in our case, at ~150 kHz for balls > 5 mm in diameter (Figure 6a). However, it is evident that the smaller steel balls predominantly contain high frequencies; for example, balls with a diameter of 0.5 mm exhibit frequencies exclusively above 100 kHz. In contrast, larger balls have a broader spectrum with a sufficient signal-to-noise ratio (SNR) from 1 kHz to 1 MHz (Figure 6a). Nonetheless, across all spectra, there is no signal with a sufficient SNR at frequencies above 1 MHz.

To correct the raw spectra and remove spectral amplitude shift caused by varying Δp across different balls, we normalize each spectrum according to its respective $\Omega_{0}$ value, which is derived from the theoretical spectrum (Table 1). However, since the 0.5 and 1 mm balls are manually dropped and the drop height is poorly constrained, the determination of the theoretical $\Omega_{0}$ is not feasible without introducing considerable uncertainty. Therefore, for these ball sizes, we manually offset the raw spectra to match the instrument apparatus response for $f<f_{c}$.

After correcting for $\Omega_{0}$, all spectra collapse into a single curve for frequencies lower than $f_{c}$, defining the instrument apparatus response I(ω) (black line Figure 6b). To compute the instrument apparatus response (I(ω)), we calculate the median of the corrected spectra in the portions where $f<f_{c}$ for each ball, considering that each ball has a different $f_{c}$.

### 4.4. Instrument Apparatus Responses and Effect of Normal Stress and Gouge on Sensor Sensitivity

The calibration function allows us to retrieve the operational modes of the sensors (Figure 7).

Specifically, spectral amplitude slopes of 0, 20, and 40 dB/decade indicate the sensor measuring displacement, velocity, and acceleration, respectively [35,47].

At low frequencies (<20 kHz), the main differences between the sensors are evident (Figure 7). The axial sensor (P) is significantly more sensitive than the transversal (S) sensors, with the vertically oriented (S_v_) sensor being more sensitive than the horizontally oriented sensor (S_h_). The greater sensitivity of S_v_ correlates with the radiation pattern of the ball drop generated on the top surface of the block.

The P sensor operates as a displacement sensor up to 30 kHz and as an accelerometer from 30 kHz to 400 kHz (Figure 7a,b). In contrast, the S sensors operate as velocimeters up to 20 kHz and as accelerometers from 20 kHz to 100 kHz; beyond this range, their response flattens, indicating they are measuring displacement.

Variations in applied normal stress, ranging from 12 to 50 MPa, show no significant effect on the calibration function for all sensors (Figure 7). It is important to note that in our setup, changes in normal stress do not affect the sensor–block coupling differently than in a triaxial configuration, where the coupling is governed by the confining pressure, influencing the sensors’ sensitivity [35]. The consistency of the instrument’s response within this normal stress range also allows us to apply the derived calibration function to different stresses.

To assess the impact of the gouge on the instrument apparatus response, we derived the calibration function using AEs generated by dropping a ball on the same block as the tested sensors and the opposite block. Analyzing the sensor on the same block as the ball drop excludes the effects of the gouge layer on the recorded waveform. Conversely, when analyzing the AE on the block opposite the ball drop, the recorded AE passes through the gouge layer, incorporating its attenuation effects into the waveform.

The instrument apparatus responses for these two scenarios are shown in Figure 8. The presence of gouge does not significantly alter the overall response of the instrument apparatus but does lead to a decrease in sensitivity due to the attenuation of the AE. For the P sensor, the gouge reduces sensitivity by approximately 5 dB across the entire frequency range (Figure 8a,b). For the S sensors (Figure 8c–f)—both S_v_ and S_h_—there is no decrease in sensitivity up to 30 kHz. Beyond this range, a noticeable decrease in sensitivity occurs at higher frequencies, reaching 10 dB at frequencies lower than 100 kHz and 20 dB at frequencies greater than 100 kHz.

### 4.5. Source Spectrum Derivation

Given the instrument apparatus response (I(ω)) and the spectrum of the raw recorded output (S(ω)), the source spectrum of the ball drops can be estimated, assuming that it is theoretically unknown (procedure indicated by the blue arrows in Figure 2). Following Equation (Equation 3), which operates in the frequency domain, we normalize the raw spectrum (S(ω)) according to I(ω) to obtain the source spectrum. Since we convert the spectral amplitude to decibels, corresponding to a logarithmic scale, the normalization simplifies to subtraction.

In Figure 9, the estimated source displacement spectra are shown superimposed with the theoretical source spectra derived from Hertz theory. The consistency of the derived source spectra with the theoretical models validates our calibration, ensuring that all instrument and path effects are accounted for in the instrument apparatus response and are removed from the raw signal.

## 5. Test Experiment

### 5.1. Experimental Setup

The test experiment is performed in a double direct shear configuration using a 3 mm quartz gouge layer (Min-U-Sil 10) with an average grain size of 3.4 µm. The experiment is performed under 100% humidity conditions, applying a constant loading rate of 10 µm/s. To modulate the machine stiffness and obtain stick-slip events (i.e., lab-quakes), we use a spring in series with the vertical piston (for details on the experimental procedure, see [48,49,50,51]).

We apply a constant normal stress of 20 MPa for the entire experiment. The experimental procedure consists of compacting the gouge for 20 min under the target normal stress of 20 MPa, followed by shearing at a constant load-point rate of 10 µm/s for 2.7 cm of vertical displacement (Figure 10). The stress displacement curve shows a first linear loading until 0.6 of friction, followed by a rollover and the beginning of small instabilities after 4 mm of fault displacement. While accumulating shear strain, the instabilities become larger (lower inset Figure 10), showing regular stick-slip behavior [51].

After 23 mm of fault displacement, fast and audible instabilities are observed, and at this point, we start to continuously record the acoustic signal with a sampling frequency of 3.125 MHz. To constrain the velocity of the S wave in the gouge, we use a survey of ultrasonic pulses in a separate experiment. By using a cross-correlation method [52], we estimate the S-wave velocity in the gouge as β=1300m/s.

### 5.2. Acoustic Emission Waveform during Lab Quake

For each stress-drop event, we record the AEs using the P and S sensors for each block. Based on the differences of the first arrival time, we can determine from which layer the AE originates, allowing us to analyze the AE waveforms using the three sensors placed on the same block where they were generated (Figure 1b). In our experiment, AE activity is detected only during the co-seismic phase that is associated with the stress drop. The seismic signals we record in this phase are also affected by low-frequency signals generated by the deformation of the forcing block during the stress drop (Figure 11b and Figure 12a).

During the co-seismic stage, when the stress drops, the gouge layer quickly compacts due to fast slip (i.e., 2–3 ms). Consequently, the response of the servo-valve that controls the horizontal piston cannot respond as fast as gouge compaction, causing a small drop in normal stress that is recovered within about 2 ms (Figure 11a). This small and rapid change in normal stress leads to the slight deformation of the side steel block. This deformation causes a low-frequency signal in the acoustic data below 10 kHz (Figure 11c). For the sake of clarity, the apparent occurrence of this signal before compaction (Figure 11c) is due to the low temporal resolution of the wavelet transform at low frequencies.

However, the AEs from the gouge occur at much higher frequencies (>100 kHz) than the signal caused by piston movement. This clear frequency separation between block deformation and AEs allows us to isolate the AEs by applying a high-pass filter to the waveform at 10 kHz (Figure 11d,e).

The horizontal piston’s effect is much more pronounced on the P sensor for two main reasons. first, the P sensor is aligned parallel to the horizontal piston, making it ideal for observing the horizontal deformations related to the normal stress drop; secondly, the P sensor acts as a displacement sensor at frequencies below 10 kHz, while the S sensor functions as a velocimeter in this frequency range (Figure 7).

The differences in the waveforms of the same AE recorded with different sensors are shown in Figure 12a–c. It is evident that the P sensor is much more contaminated by low-frequency signals than the S sensors. However, when the waveforms are high-pass-filtered, the resulting signals from all three sensors are very similar in both shape and amplitude (Figure 12d–f).

### 5.3. Source Parameter Derivation

To obtain the source spectrum of an AE, we normalize the raw spectra (S(ω)) according to the instrument apparatus response. In Figure 12g–i, we report the raw spectra of the high-pass-filtered waveform at 10 kHz (Figure 12d–f) of the P, S_v_, and S_h_ sensors for the same AE. Therefore, since F(ω)=Δp for $f<f_{c}$, we retrieve I(ω) by applying Equation (Equation 3) and using the theoretically derived Δp value (Table 1). The I(ω) values for the different sensors are reported in red in Figure 12g–i. Then, to retrieve the source spectrum, we remove the instrument apparatus response from the raw spectrum by dividing I(ω) by S(ω) (subtraction if operating in the decibel range). In the derived source spectrum, $\Omega_{0}$ corresponds to the apparent change in momentum (Δp) of the acoustic emission and is measured in units of Ns. According to [35], we use the scale factor (CFM) to convert the apparent change in momentum (Δp) obtained from spectrum calibration into moment ($M_{0}$), expressed in Newton meters (Nm). This factor (CFM = $M_{0}/\Delta p$) is empirically found to be approximately twice the wave velocity in the medium [35]. Due to the close proximity of the sensors to the source and their limited spacing, our configuration precludes the separation of P and S waves (near-field regime) that are not distinguishable in the recorded waveforms. Accordingly, we calculate the CFM based on average wave velocities in steel ($V_{p}\approx 6.1\;\mathrm{km}/s$ and $V_{s}\approx 3.5\;\mathrm{km}/s$) of 9.6 km/s in the medium. Therefore, we multiply the apparent change in momentum by the CFM to obtain the seismic moment $M_{0}$, expressed in Nm (Figure 12j–l). The resulting source spectra are reported in Figure 12j–l for each sensor.

The earthquake source spectra across a wide range of magnitudes can be represented by a scale-independent functional shape [23]. Described by the omega-squared model [53], laboratory AEs such as earthquake waveforms exhibit a flat source spectrum at low frequencies (proportional to $M_{0}$) that evolves into a high-frequency decay of $\omega^{-n}$ (with n=2 for most earthquakes). This transition is defined by the corner frequency ($f_{c}$), which is related to the source duration and inversely proportional to the source radius. Thus, to determine $f_{c}$ using a non-linear least squares optimization, we fit the source spectrum with the Brune omega-squared model [53], assuming a high-frequency decay (*n*) that can vary between 1.5 and 5 as follows:$$M\left(f\right)=\frac{M_{0}}{1+{\left(\frac{f}{f_{c}}\right)}^{n}}$$
where M(f) is the spectral seismic moment as a function of the frequency and $M_{0}$ is the total released seismic moment. We allow the high-frequency decay (*n*) to vary between 1.5 and 5 to obtain a more accurate representation of the source and to account for the correlation between this decay and event size, with smaller sources exhibiting faster decay rates [31,54]. The Brune fit and $f_{c}$ for the source spectra of the P, S_v_, and S_h_ sensors are reported in Figure 12j, Figure 12k, and Figure 12l, respectively.

Moreover, the application of circular crack-based models [23,53,55,56,57] that describe the seismic source as a circular fault patch on which stress is uniformly reduced during an earthquake allows for the estimation of the source size and stress drop, providing hints about the source physics. Following this approach and according to [53], which assumes that a rupture occurs over the radius (*r*) of a circular crack and propagates instantaneously at a constant rupture velocity of Vr=Vs, we estimate the source radius (*r*) as follows:$$r=\frac{k\beta }{2\pi f_{c}}$$
where *k* is 2.34 and β is the shear wave velocity. To estimate the seismic stress drop, we follow Eshelby (1957) [58] as follows:$$\Delta \sigma =\frac{7}{16}\frac{M_{0}}{r^{3}}$$

From the seismic moment ($M_{0}$), we estimate the magnitude following Aki and Richards (2002) [59] as follows:$$M_{w}=\frac{\log_{10}\left(M_{0}\right)-9.1}{1.5}$$

It is worth noting the consistency of the source spectrum for the same AE derived from different sensors, showing a variability in $f_{c}$ of about 20 kHz (from 70 kHz to 90 kHz) and a variability in $M_{0}$ of about 0.15 Nm (from 0.3 Nm to 0.55 Nm) (Figure 12j–l).

The analyzed AEs span a the magnitude range of −7.05 to −6.32. The largest AE has a source diameter of 2.36 cm, while the smallest is 0.28 cm. The source parameter estimates (Figure 13) obtained using various sensors show consistency, with only minor variations likely due to different attenuation effects for each sensor. It is important to note that all source sizes are smaller than the fault surface, which measures 5 × 5 cm.

The seismic stress drops for these events range from 0.075 MPa to 4.29 MPa, consistent with earthquake scaling relationships [20]. However, the mechanical stress drops range from 2 to 3.5 MPa (Figure 10). While these two types of stress drop measurements may seem similar, they should not be directly compared because the seismic stress drop relates to the fault patch that fails and produces the AE, whereas the mechanical stress drop reflects the overall stress change across the two fault gouge layers during co-sesismic slip. Our source parameter estimates are consistent with the earthquake scaling relation reported in previous studies [16,21,24,26]. This confirms that the estimated stress drops for small AEs scale with those of large earthquakes, suggesting the stress drop as a scale-invariant property. Nevertheless, in laboratory settings, as in nature, the variability in estimated stress drop spans at least two orders of magnitude, highlighting the uncertainties derived from the spectral estimates of the source parameters [23].

The consistency of our source parameters with previous laboratory studies confirms the reliability of our measurement approach and supports the validity of our acoustic system calibration.

## 6. Applicability and Limits

The empirical calibration technique reported in this article outlines a method to characterize and remove the instrument apparatus response from the AE spectrum. Despite its effectiveness, several limitations must be considered. The first involves the signal-to-noise ratio (SNR), as our spectral analyses only consider parts of the spectrum where the SNR exceeds 10 dB. High-amplitude AEs show sufficient SNR values across a broad frequency range, allowing for proper characterization of the corner frequency ($f_{c}$) and subsequent high-frequency decay. In contrast, smaller AEs have an SNR greater than 10 dB only at higher frequencies, resulting in a limited flat portion of the spectrum where the corner frequency exceeds the upper boundary of the estimated instrument apparatus. In these cases, only the seismic moment can be estimated.

A key element in our experiments is the presence of gouge layers, whose physics we aim to describe. Their presence inevitably causes the attenuation of waveform propagation. Within our experimental setup, we can identify which gouge layer generates the AE because the signals traversing the gouge layers and the central block show a distinct difference in arrival times. The time delay ranges from 25 to 50 points (around 8 to 16 µs at a sampling rate of 3.125 MHz) and is associated with a significant reduction in amplitude. The calibration of the AE is performed using the sensors located on the same block as the AE source. However, due to the small sample size, we employ (<3 mm in thickness), we cannot locate the AEs in the direction perpendicular to the fault surface. Consequently, it is not possible to account for the attenuation that the AE undergoes within the gouge layer where it is generated.

Analyzing the response of the instrumentation that accounts for the effect of the gouge (shown in Figure 8), we observe that attenuation occurs at frequencies above 30 kHz. The majority of corrected AEs in the experimental tests (Figure 13) have corner frequencies ranging from 50 to 100 kHz, with seismic moments estimated using the flat spectrum portion below 100 kHz. Below this threshold, the difference between the instrument responses with and without gouge is approximately 10 dB (Figure 8), resulting in a magnitude discrepancy of about 0.3 Mw. For smaller AEs where the seismic moment is estimated at frequencies above 100 kHz, a 20 dB difference corresponds to a magnitude difference of approximately 0.6 Mw. This highlights the importance of accurately discerning the gouge layer that generates the AE.

The estimation of the seismic moment using the eGF method is affected by radiation pattern effects, and our configuration is also subject to this issue. When we retrieve the calibration function through ball drops performed on top of the blocks, the radiation pattern results in varying sensitivities depending on the sensor orientation, with the S_v_ sensor being more sensitive than the S_h_ sensor. Furthermore, we apply the calibration to AEs generated in the gouge layer without knowing the angle between the source and the sensors and the AE radiation pattern. To mitigate this problem, it is common practice to average the spectral amplitude of the AEs across multiple sensors to smooth out amplitude differences between sensors for the same AE [21]. However, in our setup, the small sample size limits our ability to place many sensors uniformly around the fault to achieve good spatial coverage. To account for differences in seismic moment estimation due to wave radiation pattern, we estimate the seismic moment of the AEs independently for each sensor orientation (P, S_v_, and S_h_). However, we do not observe substantial differences between seismic moments estimated with different sensors (Figure 13). This lack of strong variability can be attributed to the vertical shear direction applied during the experiment, which likely induces a vertical radiation pattern in the AEs similar to that observed in the steel ball-drop experiments.

Another characteristic of the PZT to consider when characterizing the sensor response is the aperture effect. This effect consists of a decrease in amplitude response when waves have a shorter wavelength than the sensor diameter because multiple waves are averaged inside the sensor. The sensors we use have a diameter of 5 mm, meaning that, considering an S-wave velocity on steel of 3.5 km/s for waves parallel to the sensor surface, the aperture effect occurs at frequencies higher than 700 kHz. Therefore, this is well above the frequency range of our calibration.

Besides these limitations, the calibration technique we report has the strength of being carried out under experimental conditions as similar as possible to those of the typical double-direct shear experiment. Therefore, it can effectively quantify all effects (instrumental apparatus response) that change the signal from the source to the output.

## 7. Conclusions

Fault gouge experiments in a double direct shear configuration have long provided insights into the seismic mechanisms underlying earthquakes. The use of passive acoustic AE monitoring in these experiments, due to their similarity to natural earthquakes, has generated significant enthusiasm for the employment of acoustic techniques to understand the mechanisms leading to earthquakes in gouge materials [7,10,13]. However, the physical interpretation of these acoustic emissions has been limited by the absence of a calibrated acoustic system, which is essential for distinguishing the source characteristics of the acoustic emissions from their instrumental components.

The empirical calibration presented here enables the comprehensive characterization and isolation of the instrument apparatus response, which includes sensor response, instrument effects, and path effects, in a single measure. This calibration is validated through direct application to AEs generated during stick-slip events in a double direct shear experiment using quartz gouge as simulated fault gouge.

The results confirm the reliability of the proposed calibration method, demonstrating that the instrument apparatus response is accurately determined, then removed from the raw AE spectrum. This approach allows for the derivation of the source spectrum. Spectral inversion using the Brune omega-square earthquake model enables the derivation of the source parameters, showing that the source sizes and stress drops obtained from the laboratory AEs are consistent with those observed in natural earthquakes. This observation supports the idea of stress drop as a scale-invariant property [16,21,24,26].

This work provides a robust framework for the absolute calibration of AE systems in laboratory fault gouge experiments, enhancing our ability to interpret the physical processes of seismic nucleation and rupture. Future research will focus on exploring its applicability to different gouge materials to reveal different deformation mechanisms leading to fault instability and further improve our understanding of earthquake mechanics.

## Figures and Tables

**Figure 1 sensors-24-05824-f001:**
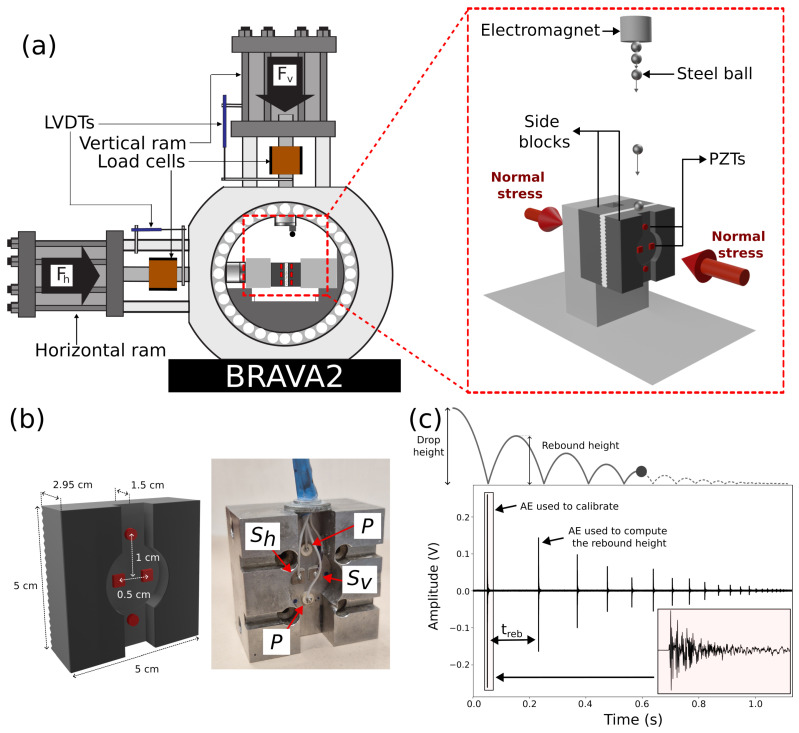
(**a**) BRAVA2 biaxial apparatus and calibration setup showing the ball-drop procedure. (**b**) Rear view of the side blocks showing the piezoelectric sensor positions. (**c**) AEs from a single ball drop, with an illustration at the top showing the ball trajectory over time. $T_{reb}$ represents the rebound time used to calculate the rebound height ($h_{reb}$). The inset shows a close-up view of the first AE used for calibration.

**Figure 2 sensors-24-05824-f002:**
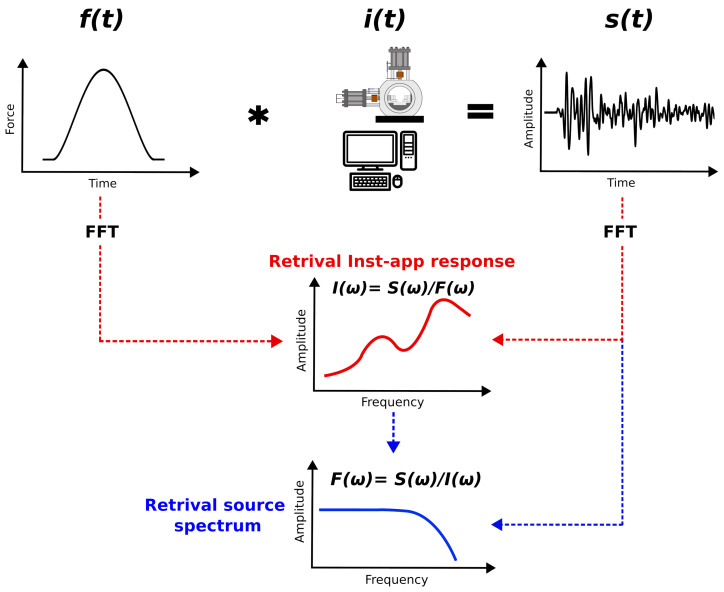
Block diagram showing the physical processes involved in signal modification and the procedure employed to derive the source spectra of AEs by removing the instrument apparatus response. From the top: the convolution (*) of the source function (f(t)) with the instrument apparatus response (i(t)) results in the recorded output signal (s(t)). In the frequency domain, from the output signal (S(ω)) and the theoretical force–time function (F(ω)), the instrument apparatus response function (I(ω)) can be characterized as indicated (red arrows). The source spectrum of an AE can then be obtained from S(ω) and I(ω) (blue arrows). FFT stands for fast Fourier transform.

**Figure 3 sensors-24-05824-f003:**
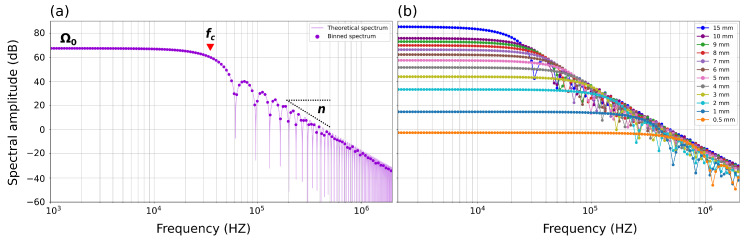
(**a**) Theoretical spectrum obtained from the FFT of the theoretical force–time function (violet shaded line) for a 7 mm ball diameter, where $\Omega_{0}$ is the spectrum amplitude of the flat portion of the spectrum, $f_{c}$ is the corner frequency, and *n* is the high-frequency decay. The violet points represent the median value for each bin (see Section 4.3 for the binning procedure). (**b**) Binned source spectra of the theoretical force–time function for all ball sizes ranging from 0.5 mm to 15 mm.

**Figure 4 sensors-24-05824-f004:**
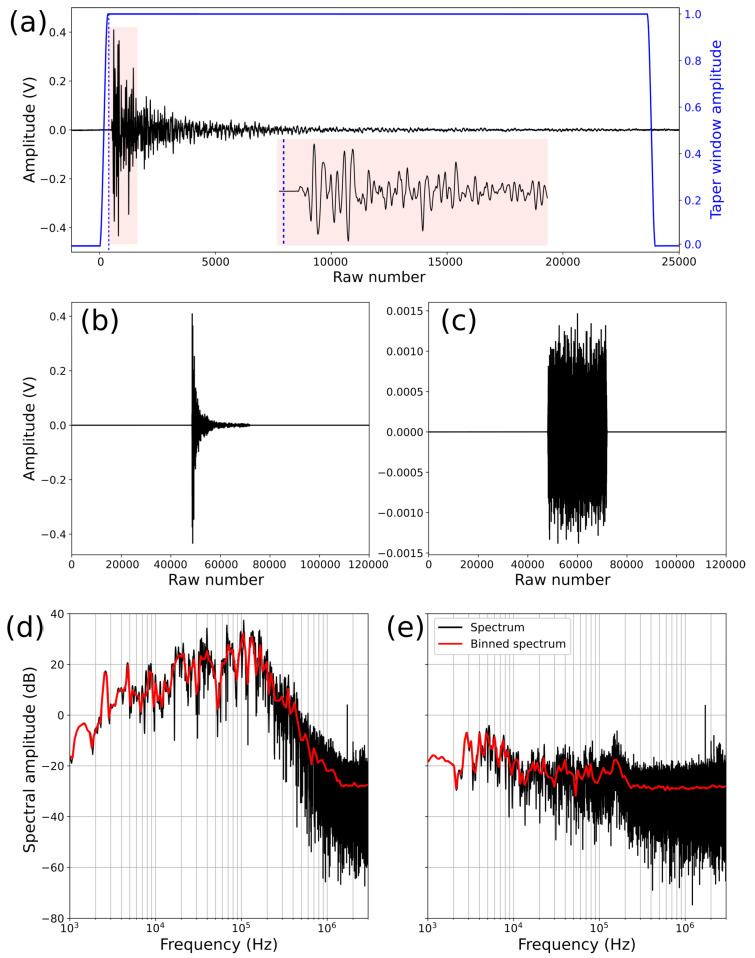
Calibration analysis for a ball drop with a 10 mm diameter recorded with the Sh sensor. (**a**) Windowing procedure for one AE, with the blue line representing the Tuckey taper window and the blue vertical dashed line representing the point where the taper window reaches 1. A zoomed-in view of the first µs is shown in the inset. (**b**) Acoustic signal after tapering and zero padding of the AE. (**c**) Acoustic signal after tapering and zero padding of the noise before the AE. (**d**) Fourier spectrum of the AE. The raw spectrum is shown in black, and the binned spectrum is represented in red. (**e**) Fourier spectrum of the noise. The raw spectrum is shown in black, and the binned spectrum is indicated in red.

**Figure 5 sensors-24-05824-f005:**
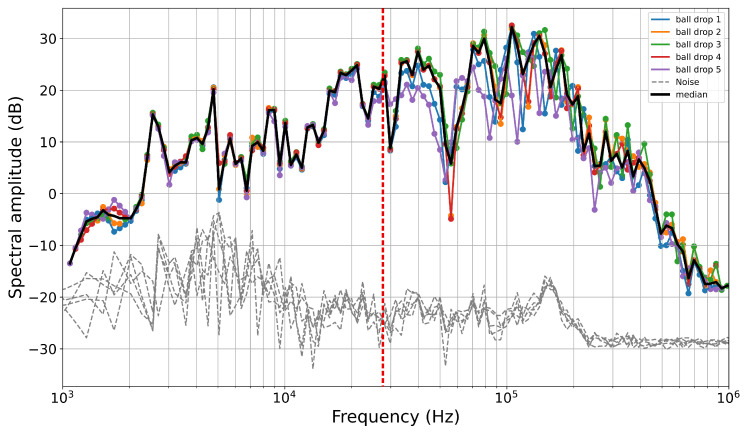
Raw spectra for five ball drops using a ball with a diameter of 10 mm. The red dashed line indicates the corner frequency of 27.6 kHz derived from Hertzian theory. To retrieve the calibration function, only the portion where $f<f_{c}$ is considered, while $f>f_{c}$ represents the high-frequency decay and is not considered in the calculation of the instrument apparatus response.

**Figure 6 sensors-24-05824-f006:**
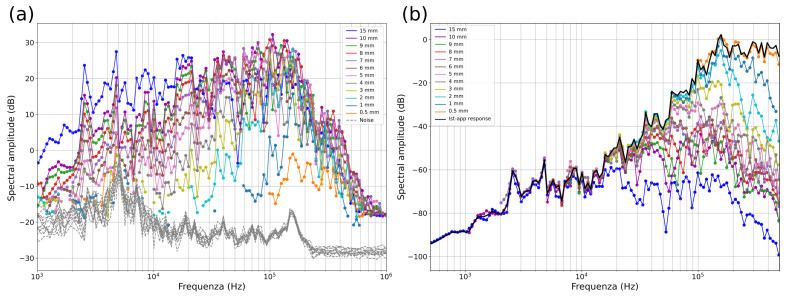
(**a**) Raw spectra of all ball drops with an SNR > 10 dB. Each line represents the median of the five ball drops performed for each ball. The gray lines represent the noise associated with each spectrum. (**b**) Spectra corrected according to the $\Omega_{0}$ value derived from the theoretical spectra. The instrument apparatus response is represented by the thick black line.

**Figure 7 sensors-24-05824-f007:**
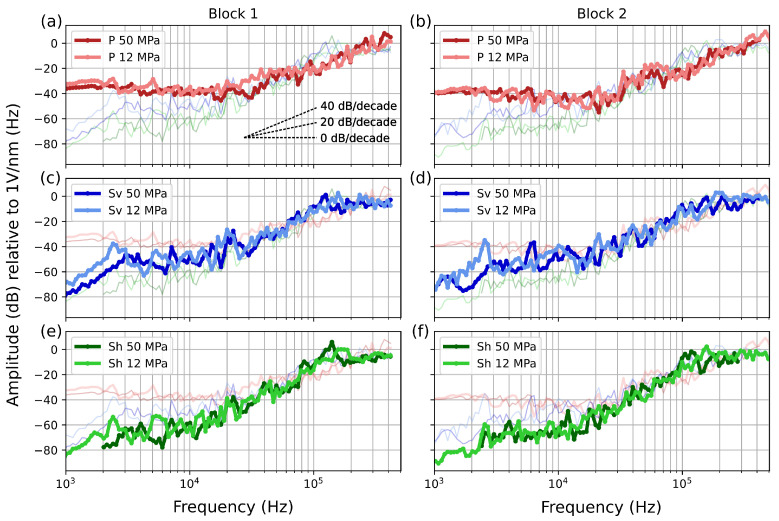
Instrument apparatus response functions for different sensors at various normal stresses and for both blocks. (**a**,**b**) The calibration function for the P sensor at 50 MPa (dark red) and 12 MPa (light red). The slopes indicate the operational mode of the sensor (0 dB/decade, displacement sensor; 20 dB/decade, velocimeter sensor; 40 dB/decade, accelerometer sensor). (**c**,**d**) The calibration function for the S_v_ sensor at 50 MPa (dark blue) and 12 MPa (light blue). (**e**,**f**) The calibration function for the S_h_ sensor at 50 MPa (dark green) and 12 MPa (light green). All calibration functions are shown in each plot to facilitate comparison.

**Figure 8 sensors-24-05824-f008:**
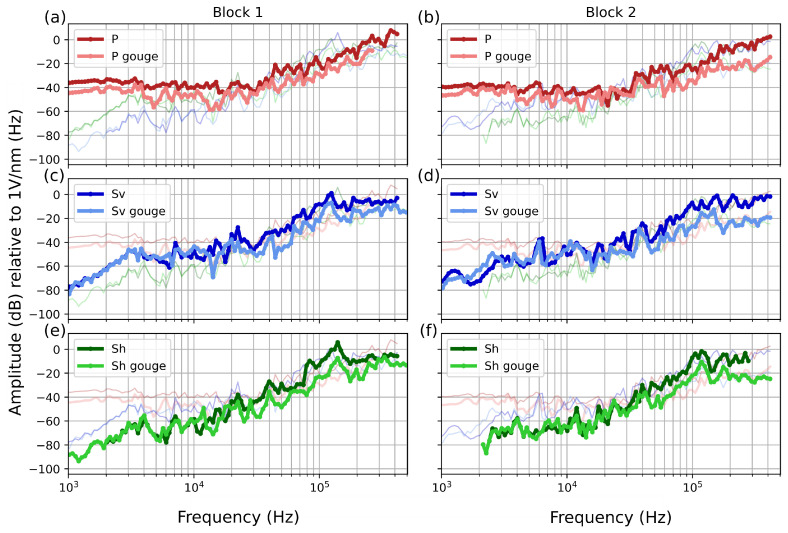
Instrument apparatus response functions for various sensors with and without accounting for the gouge layer in both blocks. Darker lines indicate the instrument apparatus response without considering the gouge layer, while lighter lines represent the response with the gouge layer considered. (**a**,**b**) The calibration functions for the P sensor. (**c**,**d**) The calibration function for the S_v_ sensor. (**e**,**f**) The calibration function for the S_h_ sensor. All calibration functions are shown in each plot to facilitate comparison.

**Figure 9 sensors-24-05824-f009:**
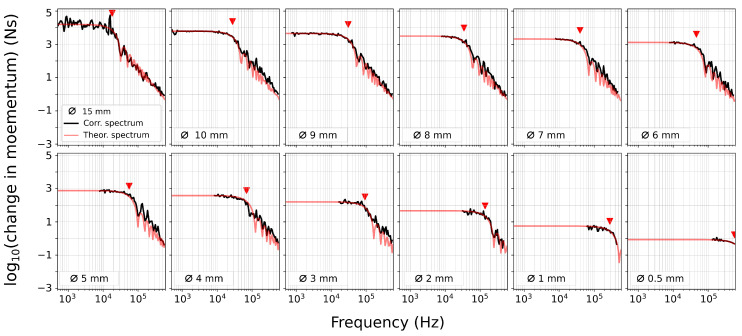
Corrected source spectra for each steel ball size with superimposed the theoretical spectra from Hertz theory. The red triangles point to the corner frequency theoretically derived from $f_{c}=1/T_{c}$. It is worth noting the slight variability in the flat spectrum for the 15 mm ball is attributed to minor variations in the impact position resulting from manual dropping.

**Figure 10 sensors-24-05824-f010:**
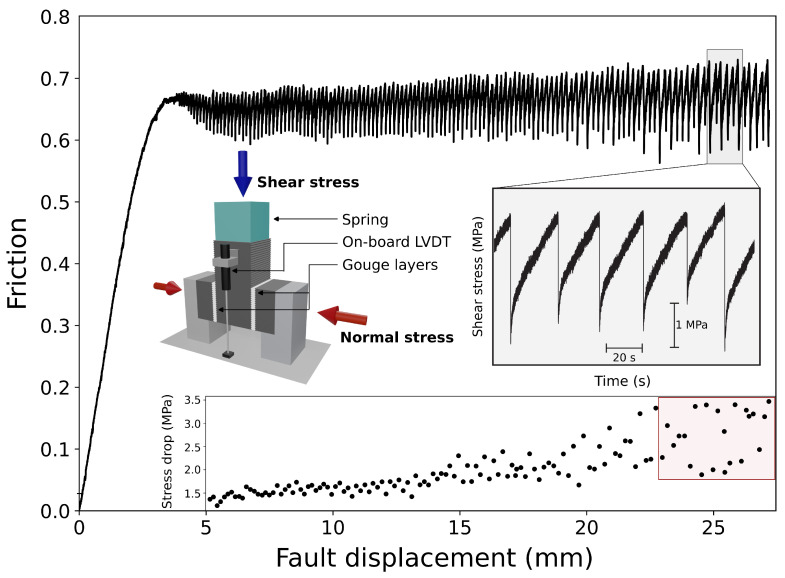
Friction curve for the test experiment. The inset on the right shows a zoomed-in view of some seismic cycles. The inset on the left shows the experimental configuration, and the inset on the bottom shows the stress drop of the events as a function of the fault displacement. The red square highlights the events analyzed with our calibration technique to estimate the source parameters.

**Figure 11 sensors-24-05824-f011:**
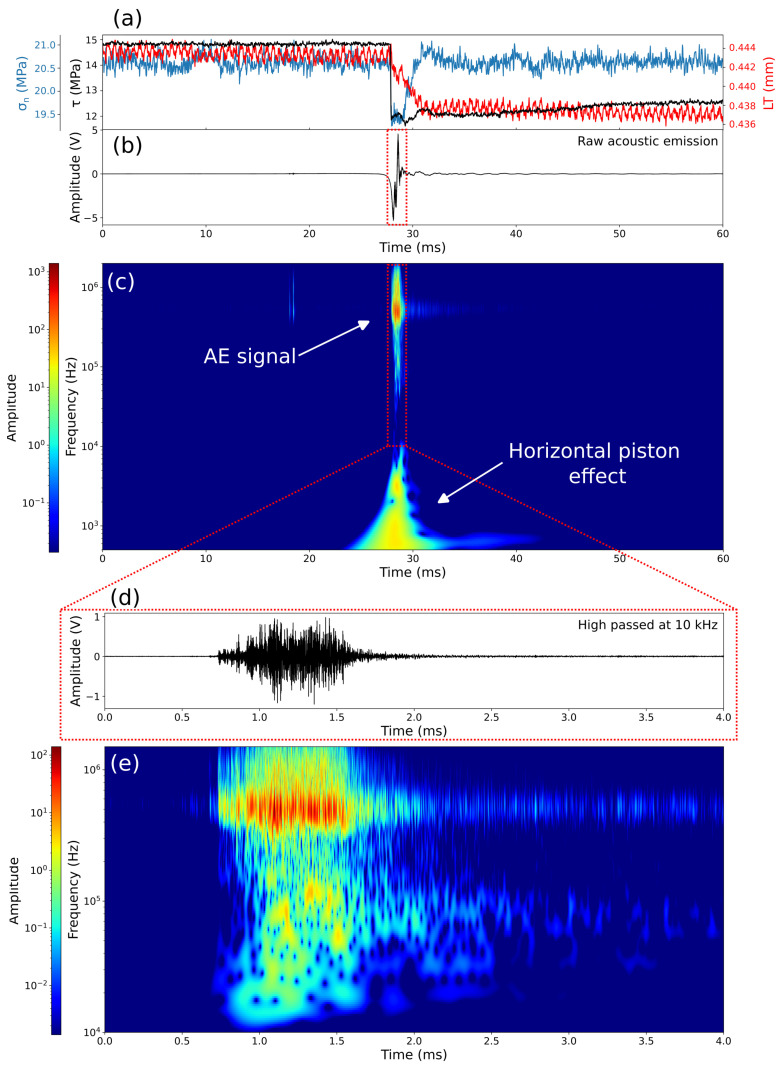
Wavelet spectrum of the seismic signal. (**a**) The evolution of shear stress (black line), normal stress (light blue line), and layer thickness (red line) for the 60 ms surrounding the stress drop. (**b**) The raw seismic signal from the P sensor. (**c**) Wavelet spectrogram of the raw acoustic signal shown in (**b**). (**d**) The high-pass-filtered waveform at 10 kHz. (**e**) Wavelet spectrogram of the high-pass-filtered acoustic signal shown in (**d**).

**Figure 12 sensors-24-05824-f012:**
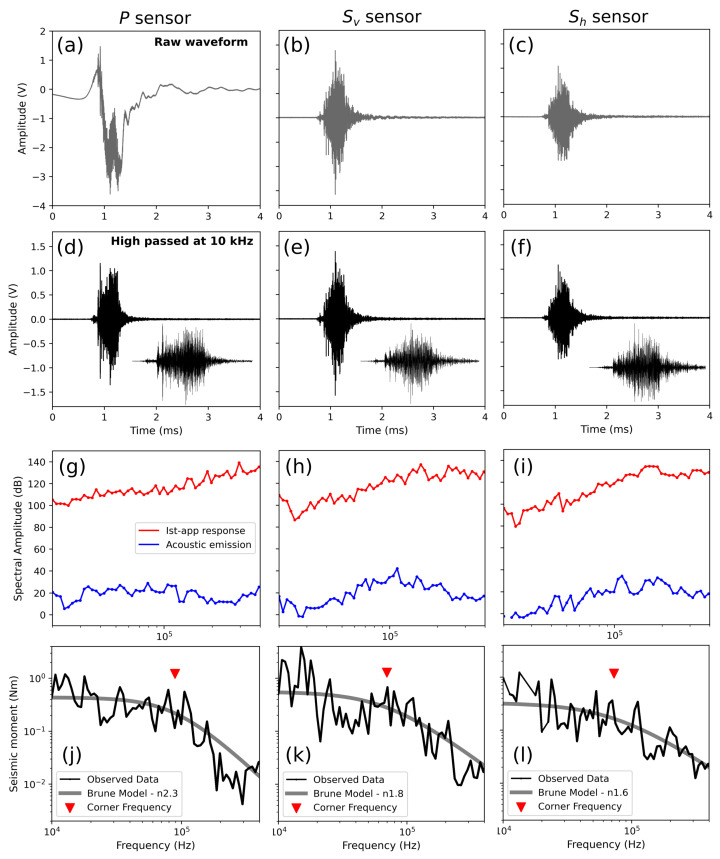
Waveforms and spectra of the same AE recorded with different sensors. The left column refers to the P sensor, the central column to the S_v_ sensor, and the right column to the S_h_ sensor. (**a**–**c**) Raw waveforms. (**d**–**f**) High-pass-filtered waveforms at 10 kHz with a zoomed-in view of the AE in the lower right. (**g**–**i**) The instrument apparatus response (red line) with the AE spectrum (blue line). (**j**–**l**) The source spectrum derived from the AE spectrum and the instrument apparatus response. The gray line represents the Brune fit, and the red triangle indicates the corner frequency.

**Figure 13 sensors-24-05824-f013:**
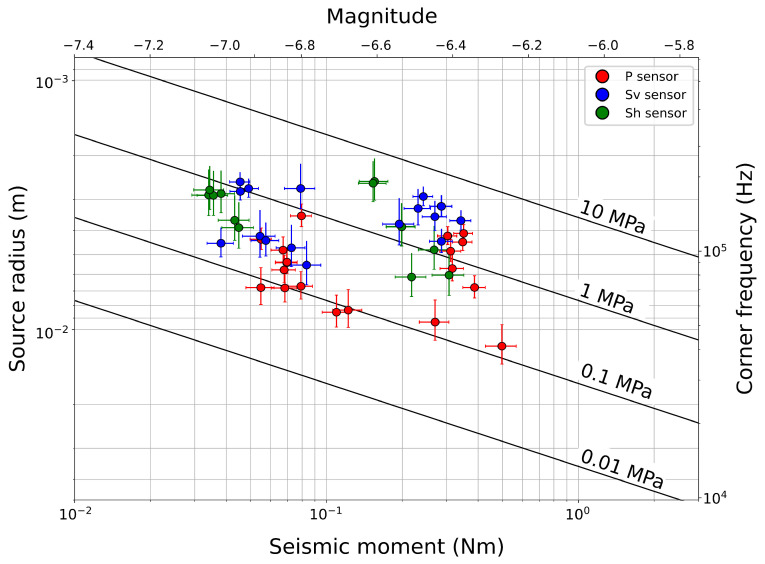
Source parameters estimated with different sensors (P in red, S_v_ in blue, and S_h_ in green). Error bars represent two standard deviations.

**Table 1 sensors-24-05824-t001:** Properties of modified lead zirconate–lead titanate (PIC 255) PZT sensors.

Density	ρ	7800 kg/m^3^
Elastic stiffness coefficient (shear direction)	$C_{55}^{E}$	$2.128\times 10^{10}$ N/m^2^
Elastic stiffness coefficient (axial direction)	$C_{33}^{E}$	$1.192\times 10^{11}$ N/m^2^
Elastic compliance coefficient	$S_{11}^{E}$	$1.606\times 10^{-11}$ m^2^/N
Poisson’s ratio	$\sigma^{E}$	0.35

**Table 2 sensors-24-05824-t002:** Properties of the steel ball drops derived from the spectra of the theoretical force–time function.

Ball Diameter (mm)	$F_{\max\mathrm{corr}}$ (N)	Δp (Ns)	$\Omega_{0}$	$f_{c}$ (kHz)
15	102.11	0.00310	18,579.44	18.237
10	51.47	0.00104	6185.67	27.608
9	41.5	0.00075	4480.6	30.702
8	32.66	0.00052	3125.33	34.569
7	24.90	0.00035	2077.93	39.542
6	18.22	0.00022	1297.28	46.171
5	12.60	0.00013	749.96	55.452
4	8.03	$6.440\times 10^{-5}$	379.72	69.373
3	4.499	$2.703\times 10^{-5}$	157.67	92.574
2	1.99	$7.971\times 10^{-6}$	46.53	138.975
1	0.49	$9.914\times 10^{-7}$	5.52	278.176
0.5	0.15	$1.486\times 10^{-7}$	0.75	561.749

## Data Availability

The collected data are available at https://zenodo.org/records/12751802 (accessed on 18 July 2024) Zenodo Record 12751802. For any further requests, please do not hesitate to contact the corresponding author at federico.pignalberi@uniroma1.it.

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
