# Peer review of "Estimating Lab-Quake Source Parameters: Spectral Inversion from a Calibrated Acoustic System"

_sensors, 2024, doi:10.3390/s24175824_

Round 1
Reviewer 1 Report
Comments and Suggestions for Authors
RE: "Estimating Lab-Quake Source Parameters: Spectral Inversion from a Calibrated Acoustic System" Pignalberi et al..
The authors do a fantastic job to describe the correct methodology to calibrate AE systems in experimental machines. This research gives an excellent overview of all aspects that need to be considered and cite the appropriate research. Some minor comments are in the attached PDF. I believe that this research is suitable for publication and only recommend to consult the comments in the PDF. There are clear limitations to this approach which the authors have addressed appropriately.

Reviewer 2 Report
Comments and Suggestions for Authors
Accept with minor corrections.

Reviewer 3 Report
Comments and Suggestions for Authors The manuscript presents a calibration technique that effectively estimates the instrument apparatus response, which includes sensor responses, instrumentation effects, and path characteristics. By calibrating the AE source spectra, the authors successfully demonstrate that the laboratory AEs exhibit stress drops and magnitudes consistent with natural earthquake scaling relations. The study can contribute to laboratory seismology by enhancing the physical insights into earthquake source process. The manuscript is well written and organized. I have several minor comments. 1. What is the most innovative contribution of this study? In other words, how the proposed technique differentiate from and advance beyond existing methods. I think most AE (and earthquake) calibration systems consider the instrument apparatus resonpse as a whole. The study also lacks necessary comparisons between the proposed calibration method and existing approaches. 2. In Figure 9, there is abnormal variations for frequency below fc for ball size r=15mm compared with others tests. This anomaly needs further explanation. 3. In Figure 12, the descriptions of subfigure (g) to (i) are missing in the caption. 4. Associated with the ratiation pattern issue discussed by the authors, the first polarity is also important for e.g., source mechanism inversion, though it might not be directly related to source spectrum. I am wondering whether the instrumental response will change the polarity of these low-magnitude AE events.Author Response
Please see the attachment.
